# Comprehensive scoping review of health research using social media data

Joanna Taylor, Claudia Pagliari

Usher Institute for Population Health Sciences and Informatics, The University of Edinburgh, Edinburgh, UK

**Correspondence to**
Dr Claudia Pagliari;
claudia.pagliari@ed.ac.uk

## ABSTRACT

**Introduction** The rising popularity of social media, since their inception around 20 years ago, has been echoed in the growth of health-related research using data derived from them. This has created a demand for literature reviews to synthesise this emerging evidence base and inform future activities. Existing reviews tend to be narrow in scope, with limited consideration of the different types of data, analytical methods and ethical issues involved. There has also been a tendency for research to be siloed within different academic communities (eg, computer science, public health), hindering knowledge translation. To address these limitations, we will undertake a comprehensive scoping review, to systematically capture the broad corpus of published, health-related research based on social media data. Here, we present the review protocol and the pilot analyses used to inform it.

**Methods** A version of Arksey and O'Malley's five-stage scoping review framework will be followed: (1) identifying the research question; (2) identifying the relevant literature; (3) selecting the studies; (4) charting the data and (5) collating, summarising and reporting the results. To inform the search strategy, we developed an inclusive list of keyword combinations related to social media, health and relevant methodologies. The frequency and variability of terms were charted over time and cross referenced with significant events, such as the advent of Twitter. Five leading health, informatics, business and cross-disciplinary databases will be searched: PubMed, Scopus, Association of Computer Machinery, Institute of Electrical and Electronics Engineers and Applied Social Sciences Index and Abstracts, alongside the Google search engine. There will be no restriction by date.

**Ethics and dissemination** The review focuses on published research in the public domain therefore no ethics approval is required. The completed review will be submitted for publication to a peer-reviewed, interdisciplinary open access journal, and conferences on public health and digital research.

### Strengths and limitations of this study

► The proposed scoping review addresses the need for a comprehensive, cross-disciplinary synthesis of health-related research using social media data.
► A five-stage scoping review protocol was developed, which was informed by an exploratory analysis of existing reviews, terminologies and published taxonomies.
► This revealed the changing usage of relevant terms over time, the types of keyword combinations most likely to yield eligible studies and the time-lag between the launch of new social media platforms and published research using data derived from them.
► The findings of the scoping review will be communicated using both static and multimedia visualisation tools.

modern day social media platform, sixdegrees, was launched in 1997 and allowed users to connect with friends and family through sending messages and posts on bulletin boards.[2] Since then the use of social media has become increasingly common[3] with an estimated 2.5 billion of the global population estimated to be using them as of 2017.[4]

As social media have become an integral part of people's lives, research focusing on social media has also evolved and is taking place in a number of domains. A bibliometric content analysis of key words that appeared in the abstracts of 14 500 journal articles and conference papers identified five main areas in which a high volume of research already exists. This includes research related to health, educational uses of social media, computing and computer science methods, business, organisational and marketing topics and political and social engagement.[5]

Uses of social media in the context of research can be described according to the broad categories identified by Bjerglund-Andersen and Söderqvist (dissemination, discussion/networking, public engagement, teaching, research/data collection).[6] Taylor and Pagliari[7] recently separated the last of these into social media as a source of data for research and as

## INTRODUCTION

Social media are online, often mobile, platforms that support the creation and exchange of user-generated content.[1] They include generic platforms for networking, information sharing and collaboration (eg, Facebook, Twitter, YouTube, Linkedin) and online forums aimed at specific communities (eg, Patientslikeme, Mumsnet). The first

**Table 1** Existing systematic, quasi-systematic or scoping reviews indexed in PubMed

| Topic | Examples of systematic, quasi-systematic or scoping reviews |
| --- | --- |
| Disease surveillance | Social media and internet-based data in global systems for public health surveillance: a systematic review.[16] |
| | Scoping review on search queries and social media for disease surveillance: a chronology of innovation.[17] |
| | Ebola and the social media.[18] |
| | Digital disease detection: a systematic review of event-based internet biosurveillance systems.[19] |
| | Utility and potential of rapid epidemic intelligence from internet-based sources.[20] |
| | Using online social networks to track a pandemic: a systematic review.[21] |
| | A systematic review of event-based public health surveillance systems.[22] |
| | Social media: a systematic review to understand the evidence and application in infodemiology.[23] |
| Adverse event monitoring | Systematic review on the prevalence, frequency and comparative value of adverse events data in social media.[26] |
| Quality of healthcare services | Social media and rating sites as tools to understanding quality of care: a scoping review.[24] |
| | Eight questions about physician-rating websites: a systematic review.[25] |
| Illicit drug use | Systematic review of surveillance by social media platforms for illicit drug use.[27] |
| eGovernment | Use of social media for e-Government in the public health sector: a systematic review of published studies.[28] |
| Chronic disease | Social media use in chronic disease: a systematic review and novel taxonomy[29] |
| Ethics | Attitudes toward the ethics of research using social media: a systematic review[30] |

a tool for the conduct of research. These 'secondary uses' of social media data include analyses of trends, associations and sentiments in users' postings, as well as interactions and networks.[8 9]

This new source of 'big data' has triggered scientific developments in a number of areas, including health. The term 'infoveillance' was coined by Eysenbach in 2009 to describe the automated harvesting and analysis of internet searches and social media postings as an alternative approach for health and disease surveillance.[10] Other terms have since been introduced across academia and industry to describe the use of such data for gleaning insights about behavioural trends, determining the impact of interventions or predicting future events. These include 'social media listening',[11] 'social media mining',[12] 'social analytics',[13] 'social machines'[14] and 'netnography',[15] to name but a few.

Based on an iterative search using PubMed, we identified an existing corpus of 15 systematic, quasi-systematic and scoping reviews on the secondary use of social media data for health research, examples of which are summarised in table 1 (see online supplementary appendix 1 for the

**Table 2** Research questions

| Aspect | List of questions |
| --- | --- |
| General | What is the total number of studies published by year? |
| | What terms are being used to describe the nature of this research? |
| | Which academic communities are most active in health research using data from social media? |
| | Where are study authors located, according to their affiliation? |
| | What is the geographical scope of the social media data analysed in these studies? |
| | For what purposes are social media data being used in this research? |
| Topic | Which health topics are being studied? |
| Social Media type | Which social media platforms or sites are being used as sources of data? |
| Extract and analysis | What units of analysis are being applied? |
| | How are data from social media being extracted and analysed and which proprietary tools are being used? |
| Ethics | How are ethical considerations applied in the published research? |

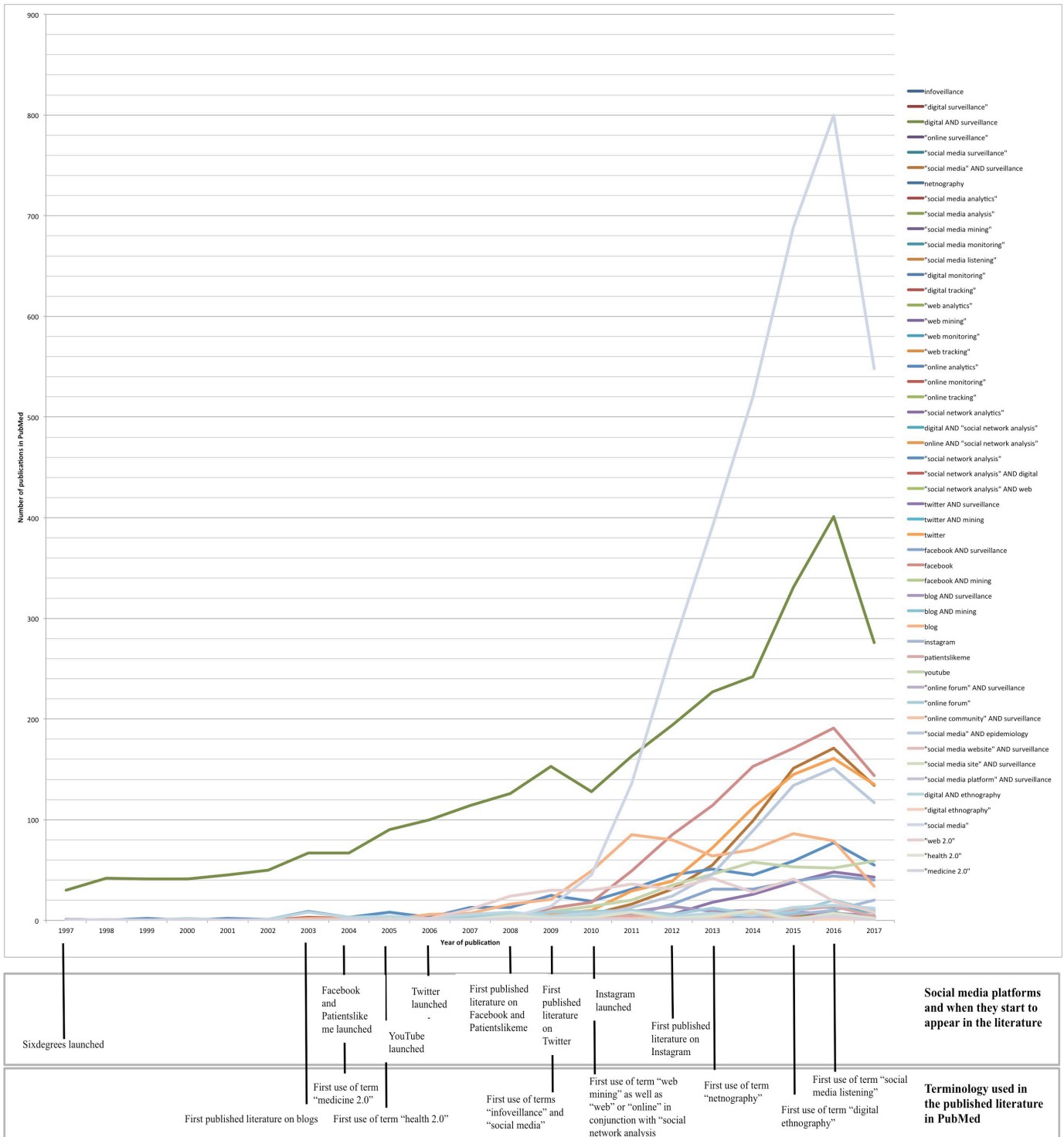

**Figure 1** Results of the search queries by year.

full list, summarised according to their aims, number of databases and findings). Among these are reviews on approaches to health surveillance[16–23] quality of healthcare services,[24 25] adverse event monitoring,[26] illicit drug use,[27] eGovernment,[28] chronic conditions[29] and ethics.[30]

A 2013 scoping review by Bernardo *et al* on the use of internet search queries and social media for disease surveillance identified the first study as has having been published in 2006 and described how techniques for exploiting this data are evolving to increase the accuracy of signal detection.[17] Although insightful in understanding the chronology of this type of innovation, the review searched only one database and focused on the surveillance of foodborne illness. A 2014 systematic review by Velasco *et al* on the impact and use of data from social media for public health surveillance went further, concluding that existing surveillance systems are limited, and there is a need for automated technologies to monitor health-related information on the internet,

> **Box 1  Article inclusion and exclusion criteria for the scoping review**
>
> **Inclusion**
> ► Types of publication: peer-reviewed research articles. Full conference papers.
> ► Language: English.
> ► Publication timeframe: 1997–2017.
> ► Types of research: empirical studies using health-related data from social media, extracted or studied in situ, using both manual and automated methods.
>
> **Exclusion**
> ► Types of publication: dissertations/theses; reports or abstracts only; letters to the editor; marketing or advertising material; reviews or editorials.
> ► Language: not English.
> ► Types of research: studies based on data from online sources other than social media (eg, internet search histories, online news reports). Commercial research aimed at obtaining market intelligence or informing product promotion. Studies examining social media platforms, rather than using them as a source of data. Studies describing social media as a communication or broadcasting channel (eg, for public health promotion).

although it did not specify which types of systems were analysed.[16] O'Shea's systematic review of event-based internet biosurveillance systems describes the wide variety of technologies and data sources for gathering, processing and disseminating data to detect infectious disease outbreaks.[19] These reviews, alongside the others listed, focus predominantly on infectious disease. A systematic review on social media for chronic disease exists[29] which focuses on understanding the clinical outcomes associated with using such technologies for patient support, education and disease management across different conditions. However, this is limited by the inclusion of only one database and did not examine the type of methods or tools used to extract and analyse social media data, the academic discipline and setting of the research or the ethical issues considered. With respect to the latter, a recent systematic review by Golder *et al*, analysed studies reporting attitudes towards the ethics of research using social media data. This revealed wide variation in attitudes, from the very positive to the very concerned, depending on the purpose and quality of the research, researcher affiliation, the potential for harm and the methods used.[30] Although it used an impressive 16 databases, this review did not examine regional, disciplinary, condition or health topic specific variations, and the authors note that the demographic characteristics of respondents were unclear in most studies. As noted in a recent review of UK research ethics guidelines and published health research, growing public awareness of the misuse of social media for marketing and algorithmic prediction are forcing policymakers to look more closely at this issue.[7]

In light of these gaps, we set out to undertake a comprehensive scoping review aimed at capturing and profiling a broad corpus of published multidisciplinary research in which data obtained from social media have been used to monitor, understand or evaluate aspects of health and disease. In this paper, we describe the formal protocol for the comprehensive scoping review, alongside the preliminary analyses undertaken to inform each stage.

## METHODS AND ANALYSIS

Scoping reviews are a type of quasi-systematic review that are increasingly used for understanding research on emerging innovations, which may be poorly indexed, distributed across published and grey literature or located in different academic disciplines.[31–33] They typically progress in five key stages (1) identifying the research question; (2) identifying the relevant studies; (3) selecting the studies; (4) charting the data and (5) collating, summarising and reporting the results. Developing an a priori review protocol can be useful for managing this complexity, while formative research can aid the design of such protocols by identifying relevant terminologies, topics and evidence sources. Scoping reviews are mainly aimed at mapping the evidence landscape rather than establishing the effectiveness of particular interventions, and typically do not involve critical appraisal of study methodology or detailed extraction of outcomes data.[34]

### Stage 1: identifying the research question
In addition to the overarching review objective articulated in the introduction, several specific questions will be used to guide our analysis of existing research evidence, as listed in table 2.

### Stage 2: identifying relevant literature
Comprehensive scoping reviews aim to capture literature from a range of electronic databases, reference lists and grey literature. As such, our approach will include:
► A systematic search of peer-reviewed studies using five health, informatics, business and cross-disciplinary electronic databases: PubMed, Scopus, Association of Computer Machinery (ACM), Institute of Electrical and Electronics Engineers (IEEE) and Applied Social Sciences Index and Abstracts (ASSIA).
► 'Snowballing' from article reference lists[35] will be used to identify additional studies that may have not been indexed in the online research databases.
► Searching grey literature from the internet using the most widely used search engine, Google.[36] The first 20 Google results yielded by each search string will be reviewed, as this search engine displays the results by relevance.

The search strategy for each of the databases was defined in consultation with a senior librarian. Our five electronic databases and one internet search engine takes account of time and funding constraints, although the sources targeted are likely to capture most of the relevant literature.[37]

#### Formative searches
Given the breadth and changing popularity of terms related to social media and social media mining, we undertook a formative analysis to understand those most likely to yield articles relevant to our review objectives. We first created an extensive list based on the search strings specified in a recent systematic review of social media in the context of e-Government in public health,[28]

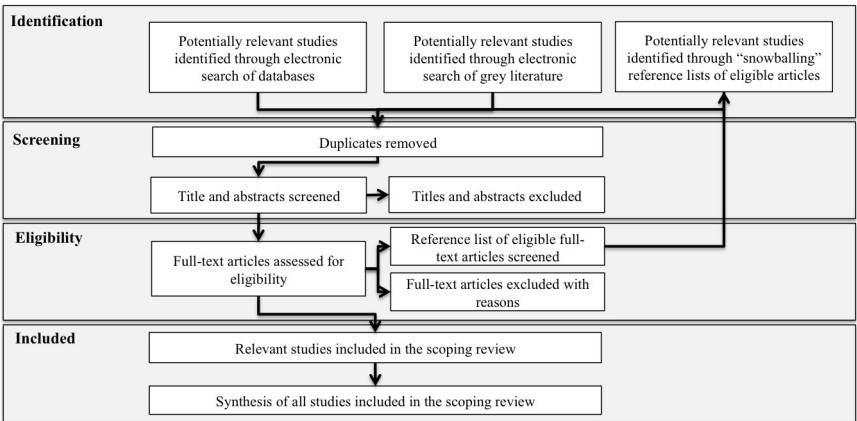

**Figure 2** Steps for identification, screening, eligibility and inclusion of studies in the scoping review.

supplemented with terms captured from iterative online searches. The list included generic terms related to social media (eg, digital, blog, social network, Web 2.0), named social media (eg, Twitter, Facebook), types of social media mining (eg, surveillance, scraping, listening, infoveillance) and analytics (eg, netnography, social network analysis) (see online supplementary appendix 2 for the full list). We then ran separate search queries for each term in PubMed, constructed as [term] AND (health OR illness OR disease) AND human. Searches yielding one or more hits were charted over time, to examine their changing frequency in the research literature. Out of the 72 terms tested in PubMed, 52 produced hits. The timeline was cross-referenced with key historical events, such as the introduction of new platforms or new methodologies. The results are shown in figure 1.

As can be seen from figure 1, 91% of relevant studies were published in the past 9 years (11 154 results between 2009–2017 and 1048 between 1997–2008). Not surprisingly, the search term 'social media' produced the highest number of results overall compared with other search queries. Regarding specific platforms, searches with sixdegrees and Linkedin produced no results, compared with other social media such as Facebook (1152 hits, between 2008 and 2017), Twitter (939 hits, between 2009 and 2017), YouTube (351 hits, between 2008 and 2017), Patientslikeme (60 hits, between 2008 and 2017) and Instagram (40 hits, between 2012 and 2017). We also observed a time lag of 4 years between when Facebook, Patientslikeme and Twitter were launched in 2004, 2004 and 2006 respectively and the first research article was published, while the equivalent latency for YouTube and Instagram was 2 years. The individual search terms 'surveillance' (665), 'epidemiology' (581) and 'ethnography' (110) produced the highest yields, compared with terms specific to digital research methods. With respect to the latter, temporal changes in the use of terms such as 'infoveillance', 'netnography', 'social media listening', 'social media analytics' and 'social media mining' indicate the evolution of innovations and research perspectives, however including such specialist terms was not critical for identifying relevant studies.

As previously noted, a systematic search of five health, informatics, business and cross-disciplinary electronic databases—PubMed, Scopus, ACM, IEEE and ASSIA—will be undertaken to identify relevant peer-reviewed studies. Below is the search query that will be used to interrogate these sources. This takes account of insights gathered during our formative analyses:

► Named social media,[38] including health-specific social media sites.[39]
► OR 'web 2.0' OR 'social media' OR 'blog' OR 'microblog' OR 'wiki' OR 'virtual world' OR 'discussion forum' OR 'online community'
► AND 'surveillance' OR 'epidemiology' OR 'infoveillance' OR 'ethnography' OR 'netnography' OR 'mining'
► AND ('health' OR 'disease' OR 'illness')
► NOT 'animal'

For Google searches the advanced search option will be used with English selected as the preferred language. All regions will be included, and the following search strings will be used:

► social media, surveillance, health, NOT animal'
► 'social media, surveillance, disease, NOT animal'
► 'social media, data mining, health, NOT animal'
► 'social media, data mining, disease, NOT animal'
► 'social media, epidemiology, health, NOT animal'
► 'social media, epidemiology, disease, NOT animal'

### Stage 3: study selection
One researcher will conduct the systematic search of electronic databases and grey literature. Studies will be selected after the abstracts and titles, identified via the electronic searches and 'snowballing' technique, have been independently screened for relevance by two researchers based on the specified inclusion criteria.

### Inclusion and exclusion criteria
Peer-reviewed journal articles and conference papers will be included where these describe empirical research using data from social media such as Twitter or Facebook, whether extracted or studied situ, using either manual or automated methods. Studies not in English, dissertations/

**Table 3** Existing classification frameworks that correspond with the research questions

| Research question | Existing classification framework to be applied | How the classification framework will be applied |
|---|---|---|
| What is the total number of studies published by year? | Not specified | The year that the eligible study was published will be captured. |
| What terms are being used to describe the nature of the research? | Not specified | The terminology used by study authors to describe the aims or methodologies used in their study will be captured for example, infoveillance, netnography, social listening. |
| Which academic communities are most active in conducting health research studies that use data from social media? | Scimago Journal Subject Areas[40] including 27 subject areas including medicine, computer science, health professions, business management and accounting as well as social sciences. | The journal in which eligible study is published, along with the affiliation of all authors as well as any sources of funding for the study (if shown) will be classified according to the disciplinary taxonomy used by the Scimago Journal ranking portal. These categories are not mutually exclusive. |
| Where is the affiliation of the first author located? | World Bank Regional and Lending Groups for Countries includes seven regions and four lending groups.[41] | The geographical location of the first author conducting the eligible study will be classified according to the regional and lending groups provided by the World Bank for 2017. These categories are not mutually exclusive. |
| What is the geographical scope of the sample of social media data analysed in the study? | World Bank Regional and Lending Groups for Countries includes seven regions and four lending groups.[41] | The geographical location of the population sample analysed within the study will be classified according to the regional and lending groups provided by the World Bank for 2017. These categories are not mutually exclusive. |
| What is the purpose for which social media data are being used in the research? | Not specified | The purpose of the eligible study will be captured. |
| Which health topics are being studied? | The 19 health-related topics which inform the WHO's Sustainable Development Goals (SDG).[42] | The type of condition and health topic being studied in the eligible study will be classified according to the list of health-related topics, which inform the WHO SDG. These categories are not mutually exclusive. |
| Which social media are used as a source of data? | Kaplan and Haenlein's eight types of social media.[1] | The type of social media from which the data for the eligible study was sourced will be classified according to Kaplan and Haenlein's eight types of social media. The name of the social media will also be captured. |
| How large are the studies and what is the unit of analysis applied? | Not specified | The sample size and unit of analysis of the eligible study will be captured. |
| How is the data from social media extracted and analysed and which proprietary tools were used? | Several types of analysis may be performed on social media data, ranging from simple descriptive statistics to qualitative research to automated real-time analytics at scale as described by Batrinca and Treleaven.[43] These types of analysis have been described in terms of computational techniques, such as natural language processing and purposive approaches such as news analytics, opinion mining, data scraping, sentiment analysis and text analytics to name but a few. | Batrinca and Treleaven taxonomy of social media analysis techniques will be used to guide our classification of the broad approaches and specific techniques demonstrated in the eligible studies. This taxonomy may be refined as a result of new insights emerging during data extraction. We will also capture the named type of analysis performed in the eligible study as well as any reference to proprietary tools used. |
| How are ethical considerations applied in the published research? | Conway's taxonomy of ethics concepts for the use of Twitter in public health surveillance and research[44] and can be applied across all manner of social media. | The application of ethical concepts in the eligible studies will be classified according to the 10 high level categories identified by Conway. These categories are not mutually exclusive. |

theses, reports or abstracts, letters to the editor and feature articles and articles intended as marketing or advertising material will be excluded. A publication time-frame of 1997–2017 will be applied. See box 1 for the inclusion and exclusion criteria for the scoping review.

Intermittent cross-checking by the second author will help to ensure the appropriate application of inclusion and exclusion criteria. In a further stage, the full-text of included articles will be independently assessed for eligibility by both authors and discrepancies resolved through discussion. Reasons for exclusion will be documented. These steps are described in figure 2.

EndNote reference management software will be used to manage the records retrieved from searches. One reviewer will independently screen the generated citations with the help of EPPI-Reviewer 4 systematic review software and undertake data extraction.

### Stage 4: charting the data

The purpose of charting data in scoping reviews is to produce a descriptive summary of the results. For this stage, we have identified existing classification frameworks that correspond with the research questions listed previously. These will form the basis of our data charting form. The classification frameworks consider the purpose for which social media data are being used in the research, the method of data extraction (including any automated data mining tools used), the analytical–interpretive approach used (including stated theoretical perspectives), the locus of the research by academic institution and geographical scope of the data, the academic discipline associated with the research and whether/how ethical issues or guidelines are considered. Each of these existing classification frameworks and taxonomies are considered in further detail in table 3 in relation to the specific research question. These classification frameworks are not intended to be totally prescriptive, and additional emerging themes will be captured throughout the forthcoming analysis. These themes will later be used to identify gaps and inconsistencies represented in the existing frameworks, for future consideration and refinement.

### Stage 5: collating, summarising and reporting the results

The extracted data will be tabulated, with rows relating to articles, columns to classification variables and cells containing the relevant information. Both frequency analysis and trend analysis will be used to chart the classified results.

### Frequency analysis

Using frequency analysis, the counts and percentages of eligible studies will be calculated. Studies will be grouped based on the classification frameworks and taxonomies applied. The result of this analysis will be a map of eligible studies represented in bubble plot, graph or tabular form.

### Trend analysis

Trend analysis will be used to present the changing frequency of research over the past 20 years, based on the aforementioned classification criteria. A map showing the characteristics of included studies will be presented using both static and multimedia visualisation tools, such as an animated bubble plot or graph.

### Public and patient involvement

No patients or members of the public were involved in the protocol design or exploratory analyses, nor do we plan to include them in the conduct of the scoping review. Our results are nevertheless likely to be of interest to citizens who use social media, and our decision to examine how researchers report ethical considerations reflects this concern. Dissemination will include accessible summaries and graphics, which we intend to make available to the public via social media.

## ETHICS AND DISSEMINATION

The completed scoping review will be submitted for publication to a peer-reviewed, interdisciplinary open access journal, in addition to conferences on public health and digital research. Findings will be presented using both static and multimedia visualisation tools.

**Acknowledgements** Thanks to Marshall Dozier for advice on the search strategy and Santosh Vijaykumar for comments on a previous draft of the manuscript.

**Contributors** JT: designed the protocol and undertook the exploratory searches, with input from CP. JT: drafted the manuscript. CP: edited the manuscript.

**Funding** JT is a self-funded PhD student at the University of Edinburgh, supervised by CP. She also works as a management consultant for Ernst and Young Switzerland although her employers are not involved in this research. CP is a collaborator on three RCUK-funded research programmes associated with this study: The Administrative Data Research Centre for Scotland, ESRC (Grant number ES/L007487/1); The Science and Practice of Social Machines EPSRC (EP/ J017728/1) and the Farr Institute for Health Informatics Research (Scotland) MRC (Grant number MR/K007017/1).

**Disclaimer** The funders had no role in study design, data collection and analysis, decision to publish or preparation of the manuscript.

**Competing interests** None declared.

**Patient consent** Not required.

**Ethics approval** Although some scoping reviews include a consultation phase, this one focuses on published research and online sources already in the public domain. No ethics approval is therefore required.

**Provenance and peer review** Not commissioned; externally peer reviewed.

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
