## [Reviewer comments · BMJ Open]

ARTICLE DETAILS

TITLE (PROVISIONAL)	A Comprehensive Scoping Review of Health Research Using Social Media Data (Protocol)
AUTHORS	Taylor, Joanna; Pagliari, Claudia

VERSION 1 – REVIEW

REVIEWER	Nina Cesare Institute for Health Metrics and Evaluation, University of Washington
REVIEW RETURNED	18-May-2018

GENERAL COMMENTS	This is a timely and needed protocol that indicates the authors have a broad yet detailed grasp on the state of social media related health research across disciplines. Anyone who has delved into this literature understands the challenge of defining its scope, understanding where and how to search for relevant content, and synthesizing its development and/or direction. These authors preface their study with a thoughtful and grounded assessment of the ways in which this literature is divided, rationales for these divisions, and explanations of how this scoping review may be designed to accommodate them. They acknowledge an often-disregarded distinction between research that uses social media as a 'source of data,' and research that uses social media for the 'conduct of research.' Likewise, the acknowledgment that social media health research is often shared in outlets not typically accessed by public health researchers - such as computer science conference proceedings - is critical to consider when drawing conclusions about the methodological and substantive state of this literature. The taxonomy of their scoping review framework employs a similar thoughtfulness. For instance, the framework includes detailed criteria regarding the extraction and analysis approach for each study. Documentation of these points helps foster future work that a.) effectively builds on past findings and b.) is replicable. I would have liked to see more discussion of this framework's limitations. The authors provide an extensive list of well-considered search queries, but what might they be missing? Their framework for extracting information from these studies is also solid, but are there facets of this work that are inconsistently represented within studies and/or difficult to quantify/document? In other words, we know what this scoping review framework tells us - what does it not tell us? Relatedly, I would have liked to see more visual and quantitative analysis of how relevant search terms change over time. The
--

	authors note that term usage varies and may impact results. But how can future scoping analyses accommodate this change? Overall, this is a useful, generalizable framework that I would suggest other digital health research turn to when assessing relevant literature. While there is room for adjustment, I would feel comfortable sharing this protocol in its current form.
--	--

REVIEWER	Christan Grant University of Oklahoma, United States of America
REVIEW RETURNED	23-May-2018

GENERAL COMMENTS	p4,¶2: What service was used to find the 14,500 journal articles. Also, 5 main areas discovered seems arbitrary. Why were these themes chosen? What which themes were excluded? If particular themes were excluded the authors should justify why. p6,L20-23: As part of the protocol, authors should note the location studied in each journal. Studies in the UK may have different outcomes compared to Australia. This trend may continue for different parts of large regions such as the west coast and east coast of USA. p6,¶1: Authors miss a significant opportunity to include papers from computer science conferences and journals. In addition to PubMed, authors should add searches for related publications with the ACM and IEEE (1,2). p6,¶4: capitalization: Patientslikeme, LinkedIn, sixdegrees p6,¶7: Authors should also exclude animals in their google search (e.g. "social media" surveillance health -animal) to remain consistent. Also, while a google search is presented, a scholar.google.com search may be more appropriate here. [1]: The ACM Digital Library https://dl.acm.org/ [2]: IEEE Xplore Digital Library https://ieeexplore.ieee.org/Xplore/home.jsp
---

VERSION 1 – AUTHOR RESPONSE

Nina Cesare (Reviewer 1):

This is a timely and needed protocol that indicates the authors have a broad yet detailed grasp on the state of social media related health research across disciplines. Anyone who has delved into this literature understands the challenge of defining its scope, understanding where and how to search for relevant content, and synthesizing its development and/or direction. These authors preface their study with a thoughtful and grounded assessment of the ways in which this literature is divided, rationales for these divisions, and explanations of how this scoping review may be designed to accommodate them. They acknowledge an often-disregarded distinction between research that uses social media as a 'source of data,' and research that uses social media for the 'conduct of research.' Likewise, the acknowledgment that social media health research is often shared in outlets not typically accessed by public health researchers - such as computer science conference proceedings - is critical to consider when drawing conclusions about the methodological and substantive state of this literature.

- Response: Thank you for your positive comments. We are pleased that you find our protocol to be thoughtful and timely given the breadth and multidisciplinary nature of research that uses data from social media. We hope that the insights garnered can be used to inform future work that further develops our collective knowledge of this field.

The taxonomy of their scoping review framework employs a similar thoughtfulness. For instance, the framework includes detailed criteria regarding the extraction and analysis approach for each study. Documentation of these points helps foster future work that a.) effectively builds on past findings and b.) is replicable.

- Response: Thank you. We are pleased that you find the review framework to be thoughtful and believe it can help to foster future work.

I would have liked to see more discussion of this framework's limitations. The authors provide an extensive list of well-considered search queries, but what might they be missing? Their framework for extracting information from these studies is also solid, but are there facets of this work that are inconsistently represented within studies and/or difficult to quantify/document? In other words, we know what this scoping review framework tells us - what does it not tell us?

- Response: Thank you for your comments. As with all classification frameworks there are limitations and the queries that have been selected for the purpose of this scoping review are not intended to be totally prescriptive. We intend to capture emerging themes throughout the forthcoming analysis, which can later be used to identify gaps and inconsistencies represented in the frameworks deployed. To emphasise this, we have added some extra text to the section labelled 'Stage 4: charting the data'. (Page 7)

Relatedly, I would have liked to see more visual and quantitative analysis of how relevant search terms change over time. The authors note that term usage varies and may impact results. But how can future scoping analyses accommodate this change?

- Response: Thank you for your recommendations. As you note, our preliminary search of using PubMed provides early insight into how terms have changed over time. In our scoping review we will visually and quantitatively map changes in terminology over time as well as across disciplines. To emphasise this, we have added a further question to Table 2 (Page 4) and Table 3 (Page 7) "What terms are being used to describe the nature of the research?"

Overall, this is a useful, generalizable framework that I would suggest other digital health research turn to when assessing relevant literature. While there is room for adjustment, I would feel comfortable sharing this protocol in its current form.

- Response: Many thanks. We are pleased that you feel comfortable sharing this protocol in its current form. We hope that by making the amendments and updates suggested this has strengthened the protocol and will further enrich our upcoming scoping review.

Christan Grant (Reviewer 2)

p4,¶2: What service was used to find the 14,500 journal articles. Also, 5 main areas discovered seems arbitrary. Why were these themes chosen? What which themes were excluded? If particular themes were excluded the authors should justify why.

- Response: Thank you for your comment. This information relates to a study conducted by Anatoliy Gruzd in 2015 and published as a blog post on www.socialmedialab.ca. The visual representation of the automated content analysis was built using VOSviewer and Web of Science. Unfortunately, we have no additional information regarding the service that was used, why the themes were chosen and which were excluded.

p6,L20-23: As part of the protocol, authors should note the location studied in each journal. Studies in the UK may have different outcomes compared to Australia. This trend may continue for different parts of large regions such as the west coast and east coast of USA.

• Response: Thank you for your recommendation. We had intended to capture this information in the research question ‘Where are these studies originating and what is their geographical scope?’ In order to ensure clarity we have separated this into two separate categories and amended the research questions (Table 2, Page 4) and classification framework (Table 3, Page 7) to capture this distinction.

p6,¶1: Authors miss a significant opportunity to include papers from computer science conferences and journals. In addition to PubMed, authors should add searches for related publications with the ACM and IEEE (1,2).

[1]: The ACM Digital Library <https://dl.acm.org/>

[2]: IEEE Xplore Digital Library <https://ieeexplore.ieee.org/Xplore/home.jsp>

• Response: Thank you for your recommendation. As we embarked on the design of this scoping review, we had much discussion about the databases that should be included in order to maximise potential coverage of this multidisciplinary research field. ACM and IEEE were included in previous versions of this protocol and after consultation with a senior librarian were later removed. However, given your recommendation, we have decided to include them in the upcoming scoping review. This addition is reflected in the Abstract and in the section labelled ‘Stage 2: identifying relevant literature’.

p6,¶4: capitalization: Patientslikeme, Linkedin, sixdegrees

• Response: Thank you. We have updated these typos in the manuscript.

p6,¶7: Authors should also exclude animals in their google search (e.g. "social media" surveillance health -animal) to remain consistent. Also, while a google search is presented, a scholar.google.com search may be more appropriate here.

• Response: Thank you for your recommendations. We have updated the Google search queries to exclude animals. This update is reflected in the section labelled ‘Stage 2: identifying relevant literature’. We were advised, by our medical librarian, to use a Google search in preference to Google Scholar, to avoid missing eligible research not appearing in the academic databases. We are aware of the limitations associated with this and intend to discuss these in the output of our scoping review.

VERSION 2 – REVIEW

REVIEWER	Nina Cesare Institute for Health Metrics and Evaluation, University of Washington
REVIEW RETURNED	15-Jul-2018

GENERAL COMMENTS	This protocol is timely and useful to any researcher who has attempted to parse through the broad and interdisciplinary literature addressing social media and health. There are many things to appreciate about this manuscript, include the authors’ attention to research ethics, their acknowledgement of the prevalence of grey and interdisciplinary literature in this field, and the breath the proposed search parameters they provide. I also appreciate that the authors acknowledge the changing nature of digital research search terms, but recognize and confirm that these ‘specialist terms’ are not necessary for identifying relevant literature.
--

	I'm happy to see the inclusion of computer science search outlets (such as the ACM digital library) within this protocol. However, this has the potential to significantly change the scope and character of the literature covered by this scoping review. While the process of literature extraction is well documented and replicable, the within-study information extraction may benefit from a little refinement to accommodate this change. Specifically, I have some reservations about the applicability of some research questions outlined under "general," however. I'm unclear what the authors mean by "how frequently...this type of research has been published in recent years" – perhaps this can be quantified in a more directly replicable way (e.g. average number of studies under this topic published per year)? For the question "what terms are being used to describe the nature of this research?" the authors may consider referencing a specific glossary - perhaps one not connected to 'specialist terms' if these are not necessary to map the literature. Also, and this is a point that may extend beyond the scope of this protocol, but it may be important to have a taxonomic way of identifying interdisciplinary work. The authors suggest noting the affiliated field for the lead author and publication outlet, and it may be worth noting which subfields seem to invite interdisciplinary collaboration – especially between formal and natural or social sciences. One minor point: The legend for the graph on 12 is a little overwhelming. I'd encourage the authors to find alternative ways to visualize search trends (perhaps by grouping lines, or faceting the chart) in a way that conveys the same key information but is easier to read.
--	--

VERSION 2 – AUTHOR RESPONSE

Nina Cesare (Reviewer 1):

This protocol is timely and useful to any researcher who has attempted to parse through the broad and interdisciplinary literature addressing social media and health. There are many things to appreciate about this manuscript, include the authors' attention to research ethics, their acknowledgement of the prevalence of grey and interdisciplinary literature in this field, and the breath the proposed search parameters they provide. I also appreciate that the authors acknowledge the changing nature of digital research search terms, but recognize and confirm that these 'specialist terms' are not necessary for identifying relevant literature.

- Response: Thank you for your additional positive comments. We are pleased that you agree with our strategy, which takes account of the need to capture the wide range of terms used to describe this type of digital research whilst also acknowledging specialist terms that are in common use.

I'm happy to see the inclusion of computer science search outlets (such as the ACM digital library) within this protocol. However, this has the potential to significantly change the scope and character of the literature covered by this scoping review. While the process of literature extraction is well documented and replicable, the within-study information extraction may benefit from a little refinement to accommodate this change.

• Response: Thank you for this observation. The inclusion of computer science databases is indeed likely to increase the volume and diversity of studies identified by our searches. The approach that we have described allows for refinement of the data extraction categories if additional themes should arise (p7 lines 2-4).

Specifically, I have some reservations about the applicability of some research questions outlined under “general,” however. I’m unclear what the authors mean by “how frequently...this type of research has been published in recent years” – perhaps this can be quantified in a more directly replicable way (e.g. average number of studies under this topic published per year)?

• Response: Thank you for seeking clarification. “How frequently this type of research has been published in recent years?” refers to the total number of eligible studies that are included in our sample within any given year (e.g. 25 eligible studies published in 2010). We have refined this research question in Table 2, to make our intention clearer.

For the question “what terms are being used to describe the nature of this research?” the authors may consider referencing a specific glossary - perhaps one not connected to ‘specialist terms’ if these are not necessary to map the literature.

• Response: Thank you for this suggestion. As the language used in this type of research is interdisciplinary and evolving, we prefer to capture the terms used by the study authors themselves, to avoid introducing interpretation bias by applying a specific glossary. This approach will enable us to describe changes over time. We have amended the relevant row in Table 3, to clarify this.

Also, and this is a point that may extend beyond the scope of this protocol, but it may be important to have a taxonomic way of identifying interdisciplinary work. The authors suggest noting the affiliated field for the lead author and publication outlet, and it may be worth noting which subfields seem to invite interdisciplinary collaboration – especially between formal and natural or social sciences.

Response: Thank you this useful suggestion. We have edited the text in Table 3 (row 3) to include capturing the affiliation of all study authors.

One minor point: The legend for the graph on 12 is a little overwhelming. I’d encourage the authors to find alternative ways to visualize search trends (perhaps by grouping lines, or faceting the chart) in a way that conveys the same key information but is easier to read.

• Response: Thank you. We appreciate that the static graph on p12 contains a lot of information. When the scoping review has been completed we intend to use a range of multi-media visualization tools to communicate the various results and we are currently identifying open access journals that can accommodate these publication requirements.

VERSION 3 – REVIEW

REVIEWER	Nina Cesare Institute for Health Metrics and Evaluation
REVIEW RETURNED	28-Aug-2018

GENERAL COMMENTS

The authors did a nice job of clarifying questions I had regarding the scope and application of their research questions. The addition of computer science literature the previous iteration of this manuscript has the potential to significantly expand the breadth of literature defined by this framework. Questions two and three play a large role in bounding this expansion. The authors' refinement of the application framework for these questions does, I believe, make this review framework more directly applicable. I think there may be some missing punctuation in the application framework for research questions two, however. The wording is a little unclear.

I still have some concerns regarding the search terms used and the way in which they might evolve over time, but it's understandably difficult to curate a stable glossary of terms given that they belong to a sort of 'folksonomy.' I appreciate that the authors did take care to map how vocabulary has evolved and note whether specialized terms are useful in finding relevant work, and I expect that the expertise of the researchers using this framework will guide them to the correct search terms.

Concerns regarding search terms aside, I think the foundational contribution of this framework – namely, that the intersection of digital data research and health research is broad, interdisciplinary, and incorporates grey literature – is captured by this protocol. I encourage publication so that others working in this area can find and synthesize the disparate literature within this subfield.